# Case Report: “Spontaneous Descemet Membrane Detachment”

**DOI:** 10.3390/jcm12010330

**Published:** 2022-12-31

**Authors:** Antonio Moramarco, Danilo Iannetta, Luca Cimino, Vito Romano, Lorenzo Gardini, Luigi Fontana

**Affiliations:** 1Ophthalmology Unit, IRCCS Azienda Ospedaliero-Universitaria di Bologna, 40138 Bologna, Italy; 2Department of Surgery, Medicine, Dentistry and Morphological Sciences, University of Modena and Reggio Emilia, 41100 Modena, Italy; 3Ocular Immunology Unit, Azienda USL-IRCCS, 42123 Reggio Emilia, Italy; 4Ophthalmology Clinic, Department of Medical and Surgical Specialties, Radiological Sciences and Public Health, University of Brescia, 25121 Brescia, Italy

**Keywords:** Descemet membrane detachment, anterior uveitis, anterior segment optical coherence tomography, sulphur hexafluoride, corneal edema

## Abstract

Introduction: We report a case and discuss the clinical characteristics and treatment of spontaneous Descemet membrane detachment (DMD). Case description: We describe a rare case of spontaneous DMD in a patient with prior anterior uveitis and provide a review of the current literature. A 20-year-old woman with a prior history of anterior uveitis presented with vision loss in the left eye. The slit-lamp examination showed corneal edema secondary to DMD, confirmed by anterior segment optical coherence tomography (AS-OCT). The patient underwent an intracameral injection of 20% sulphur hexafluoride (SF_6_) with complete resolution of the DMD. Although rare, several cases of spontaneous DMD have been reported in the literature, mostly occurring after intraocular surgery. We searched the Pubmed database (1949–2021) for peer-reviewed publications relevant to the topic of spontaneous DMD. Discussion: The pathogenesis of spontaneous DMD is complex and depends on several factors. It can occur due to anatomical anomalies, inflammatory disease, trauma, chemical injuries, and surgical or laser procedures. In most cases, early diagnosis and appropriate management led to resolution.

## 1. Introduction

Descemet membrane detachment (DMD) was first described by Weve in 1927. It is a separation of the endothelium–Descemet membrane complex from the posterior corneal stroma; most DMDs occur as an uncommon complication of intraocular surgical procedures. The Descemet membrane (DM) is tightly attached to the posterior corneal stroma by a narrow transitional zone of the amorphous extracellular matrix known as the interfacial matrix. Thus, the rupture of the DM leads to penetration of aqueous humor into the corneal stroma and subsequent localized or diffuse stromal edema [1], which can cause loss of vision with a double anterior chamber.

Spontaneous DMD has been described as an intra or post-operative complication of other ocular surgeries such as glaucoma, corneal transplantation, pars plana vitrectomy, intracorneal ring segment implantation, radial keratotomy, and cataract. The reported incidence of 2.5% during or after extracapsular cataract extraction and 0.044–0.5% during or after phacoemulsification could be underestimated because most of the DMD occurs in a subclinical fashion [1,2,3], but a higher incidence has been reported in patients evaluated using gonioscopy (up to 47%) [4]. DMD can also occur spontaneously or after trauma, chemical injuries, and laser procedures [4,5,6,7].

Risk factors for DMD include anatomic predisposition (intrinsic DM abnormalities or endothelial diseases), bleeding from corneal vascularization, intraoperative factors such as clear corneal incisions that might create lateral traction, use of blunt blades for making incisions, inadvertent insertion of instruments between the stroma and DM, entry into the anterior chamber in a soft globe, tight or small incisions as compared to the size of the probe, improper incisions such as shelved or oblique incisions, and accidental injection of substances such as saline, air, antibiotics, or ophthalmic viscosurgical devices (OVDs) between the DM and the posterior stroma.

Mackool and Holtz described DMD as planar when there is 1 mm or less of separation between the DM and the overlying stroma in all areas and nonplanar when the DM detachment exceeds 1 mm of separation [8].

Herein, we describe a rare case of spontaneous DMD in a patient with prior anterior uveitis and review the current literature.

## 2. Case Description

A 20-year-old woman with a recent (6 months) diagnosis of idiopathic chronic anterior uveitis was referred to our cornea service because of persistent central corneal edema in her left eye. The patient was being treated at another hospital for anterior uveitis, which was treated with topical steroid therapy, but after 3 months, the central corneal edema developed. At the first examination, the best-corrected visual acuity (BCVA) was 20/20 in the right eye and hand movement in the left eye. Slit-lamp examination revealed a normal anterior segment, a clear lens in the right eye, and central corneal edema with an extensive central DMD, posterior synechiae, and dense cataract in the left eye. No signs of active intraocular inflammation were visible through the paracentral cornea. The intraocular pressure was 17 mmHg in the right eye and 20 mmHg in the left eye. No iris atrophy was present. We tried to perform an endothelial cell count, but given the condition of the cornea, the machine was unable to perform it. An AS-OCT scan was performed, confirming DMD in the central cornea. A customized uveitis workup was planned. 

The patient was treated with oral 25 mg Prednisone once a day (tapering down the dose over 14 days), 125 mg oral Acetazolamide thrice a day, topical Dexamethasone four times a day (tapering down the dose over ten days), topical Timolol-dorzolamide twice a day, and a topical hypertonic saline solution four times a day. After two months, given the persistent DMD, an intracameral injection of 20% sulphur hexafluoride (SF_6_) and an inferior basal iridectomy were performed (Figure 1). Complete re-attachment was not achieved (Figure 1), so a new 20% SF_6_ injection was performed, obtaining a definite resolution of the DM at subsequent follow-up. Slit-lamp examination at three weeks follow-up showed significant fibrotic changes at the posterior corneal stroma–Descemet membrane interface, which was responsible for limited visual recovery (Figure 2) [9]. BCVA after treatment was 20/28. We performed an endothelial cell count that showed a low endothelial cell count (Figure 3).

## 3. Discussion

Although rare, several cases of spontaneous DMD have been reported in the literature, mostly occurring after intraocular surgery. Such cases are mainly related to the mechanical stress exerted on the DM, causing separation along a cleavage plane deriving from the peculiar anatomical and functional interaction between the DM and the posterior corneal stroma [1].

Various clinical features have been described in the corneas of patients suffering from anterior uveitis, including keratic precipitates, iridocorneal adhesions, and band keratopathy [9]. Corneal edema is sometimes described in the setting of acute anterior uveitis, especially when there is a significant increase in the IOP or when there is considerable loss of endothelial cells, such as in herpetic endothelitis. Acute and/or chronic intraocular inflammatory diseases may be responsible for endothelial cell dysfunction and damage, which is likely related to metabolic changes induced by inflammatory cytokines in the aqueous humor or by the deposition of keratic precipitates [10]. Animal models have described several ultra-structural modifications, such as a break in the tight junction proteins. A reduction in endothelial cell density has been described as well [9,10,11,12].

In the case described in this study, the patient was seen only several months after the diagnosis of anterior uveitis, so we cannot exclude that the DMD may be related to the traction exerted by an inflammatory membrane overlying the endothelium. However, in our opinion, this hypothesis is not strong because of the central location of the DMD; on the contrary, we could expect an inferior detachment due to the gravitational deposition of the inflammatory material. Conversely, the spontaneous DMD could represent the consequence of an acute endothelial dysfunction in a patient with an already existing predisposition to DM detachment. 

When DMD is localized, it may resolve spontaneously, but a surgical procedure is required when it is significant with severe vision impairment. However, spontaneous resolution depends not only on the size of DMD but also on the state of health of the corneal endothelium.

The lack of adherence after the first bubbling with 20% SF_6_ and the fibrotic changes of the posterior corneal stroma–DM interface could be related to the underlying inflammation.

The management options for a DMD vary from observation to medical or surgical treatment. Several studies reported a spontaneous resolution of corneal edema, most often occurring in localized and planar DMD cases. Medical therapy includes topical steroids and topical saline solution four or more times daily. A mean period for spontaneous reattachment of DMD of 9.8 weeks has been reported [13], but Assia et al. described spontaneous reattachment of DMD within 2–3 months [14]. 

It is preferred to use a surgical approach in cases of nonplanar, extended DMD, scrolled edge, and in patients that are non-responsive to the medical therapy. The gold standard treatment for DMD is descemetopexy with intracameral air/gas. When this fails, an endothelial or penetrating keratoplasty is chosen. The use of isoexpansile gas reduces the need for repeat injections, such as 12–14% perfluoropropane (C_3_F_8_) and 14–20% sulphur hexafluoride (SF_6_) [1,3,14]. A supine position is mandatory for at least one or two hours after descemetopexy. This procedure is associated with a risk of raised intraocular pressure and pupillary block glaucoma. For this reason, several surgeons prefer to perform a pre-operative laser peripheral iridotomy or intra-operative inferior peripheral iridectomy. 

A few authors described the use of 10–0 nylon sutures for transcorneal suture fixation of DMD, which is usually combined with intracameral air/gas injection. This procedure carries the risk of DM damage and tenting at the site of origin of the DMD [3]. Drainage of interface fluid is reported in combination with descemetopexy when the latter method alone fails [3].

The prognosis for reattachment depends on the configuration of the detachment, such as the endothelium health. When medical treatment and/or descemetopexy fail, keratoplasty is the last option. Most of the procedures performed are endothelial keratoplasty or PK. The type of procedure depends on DMD features. The longer the edema lasts, the more likely the corneal stroma will be damaged. In these cases, PK is preferred.

There are other causes described in the literature that could lead to DMD, such as chemical trauma, post penetration keratoplasty, and other rare causes. 

Ocular chemical injury represents 10–22% of all ocular trauma [5,6]. The severity of ocular involvement depends on the agent’s nature, duration of contact with the ocular surface, and quickness in initiating an appropriate therapeutic protocol. Few cases of DMD after chemical injuries are described in the literature, most of which are caused by alkali agents. The onset of DMD has been reported to be from two to four months after the chemical injury. Most DMDs are located in the inferior cornea and are sometimes associated with hyphema [6,7,15]. In the reported cases, treatment has been attempted using conservative medical therapy, [15] and surgery using penetrating keratoplasty (PK) in case of failure [16]. Most of the authors preferred to perform an intracameral air/gas injection [7,15,16,17,18]. In one case, this treatment was successful [17], in two cases it did not work [7,16], and in another case, the patient was lost for follow-up [15]. 

The cause of DMD after chemical injuries is not well understood. Yuen et al. [18] observed a gas bubble between the corneal stroma and DM. They hypothesized that it was caused by the breakdown of a considerable amount of hydrogen peroxide due to the relatively high concentration of hydrogen peroxide to which the patient was exposed. The gas bubble probably pushed the DM, detaching it from the corneal stroma.

Due to the presence of hyphema in their cases, Najjar et al. [7] presumed that the mechanism of DMD was an inflammatory retrocorneal membrane that pulled and detached the DM. Alternatively, a retrocorneal membrane could have developed new blood vessels that subsequently ruptured and filled the potential space between DM and the overlying stroma.

Zhang et al. [17] hypothesized a tractional mechanism due to the formation of an inflammatory adhesion between DM and the iris. The inferior location of the detachment was probably related to the gravitational deposition of fibrin and inflammatory cells.

Spontaneous DMD has been reported as a rare complication after PK. Because it is associated with graft edema, it can be misdiagnosed as acute graft rejection or late graft failure. Most of the reported cases occurred in patients with keratoconus [1,19,20,21]. Lin et al. [22] reported the only case of spontaneous DMD in recurrent pellucid marginal degeneration after PK. Late spontaneous DMD after PK may occur at any time after surgery (up to 33 years after PK). 

The exact pathogenesis of DMD after PK is not known. Gorski et al. [20] proposed two possible mechanisms: a retrocorneal membrane developing along the graft–host interface mechanically detaching the graft DM (not proven histopathologically) or in the case of keratoconus, a progression of the ectatic disease (post-PK ectasia) in the peripheral host tissue. They hypothesized that progressive changes in the host stromal collagen might mechanically pull on the graft DM, causing spontaneous DM detachment. 

Because corneal edema after PK can also be caused by rejection or failure, the diagnosis came later in many cases. The patients were first treated with topical steroids with no results. Hence, it is crucial to perform an anterior segment optical coherence tomography (AS-OCT) to exclude DMD when a graft edema is present. 

Most spontaneous DMDs after PK cases were treated with one or more intracameral air/gas injections, achieving resolution in several cases. When this treatment failed, the authors chose to perform a new PK or a Descemet’s stripping automated endothelial keratoplasty (DSAEK) procedure. 

DMD has been reported after glaucoma surgery in patients who often required multiple air/gas injections due to the passage of the gas through the drainage system [3].

The incidence of post-canaloplasty DMD is reported to be 1.1% to 1.6% [23]. The exact mechanism of DMD after canaloplasty is not understood. Palmiero et al. hypothesized that an intrinsic abnormality in stromal adherence to the DM contributes to its detachment [24].

Liu et al. [25] reported a case of spontaneous DMD after argon laser peripheral iridotomy. They thought corneal decompensation resulted from significant endothelial cell loss due to excessive heat production during the iris photo vaporization process. After two weeks, the DMD spontaneously resolved thanks to the endothelium’s spreading over the separate area, yet this hypothesis is not certain.

Rosetta et al. observed DMD in a patient 20 years after radial keratotomy [26]. The patient was successfully treated with topical 5% sodium chloride. The authors hypothesized that DMD occurring many years after radial keratotomy might result from corneal dialysis occurring through the incisions with an osmotic in-flow responsible for the DMD. Inverting the osmotic flow, the topical hypertonic solution induced a progressive absorption of the fluid between the DM and the stroma, promoting its reattachment [26].

Cases of spontaneous DMD occurred in patients with osteogenesis imperfecta [27,28], anterior megalophthalmos [29], DMD among siblings [30], and bilateral DMD in the same patient [13], which stressed the hypothesis of an anatomic predisposition. 

Felipe et al. [30] proposed the possible role of an abnormal adhesion between the stroma and DM caused by functional deficiency of the big-h3 protein (a protein with an anchoring function between the corneal stroma and DM). They concluded that spontaneous bilateral DMD might reflect a weak adhesion between the stroma and DM caused by this protein’s functional deficiency. The familial tendency may indicate a hereditary abnormality or dysfunction of the DM–endothelium complex.

In conclusion, Descemet’s detachments are rare clinical conditions that can be correctly diagnosed by the clinical features and imaging methods such as AS-OCT; the gold standard treatment for DMD is descemetopexy with intracameral air/gas, but when this fails, endothelial or penetrating keratoplasty is chosen.

## Figures and Tables

**Figure 1 jcm-12-00330-f001:**
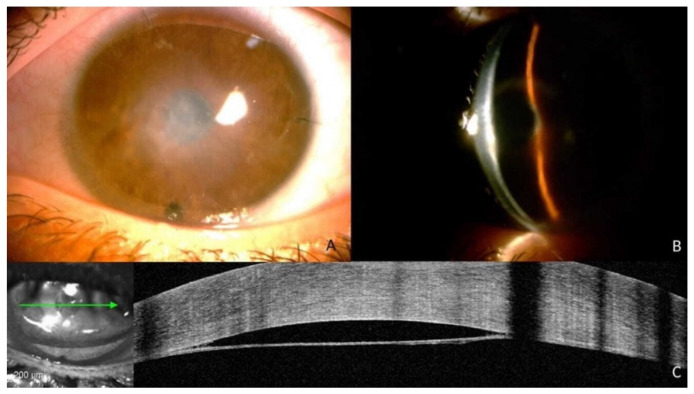
(**A**,**B**) Slit-lamp image showing persistent central corneal edema after the first injection of SF_6_; (**C**) AS-OCT shows a large Descemet membrane detachment in the central cornea.

**Figure 2 jcm-12-00330-f002:**
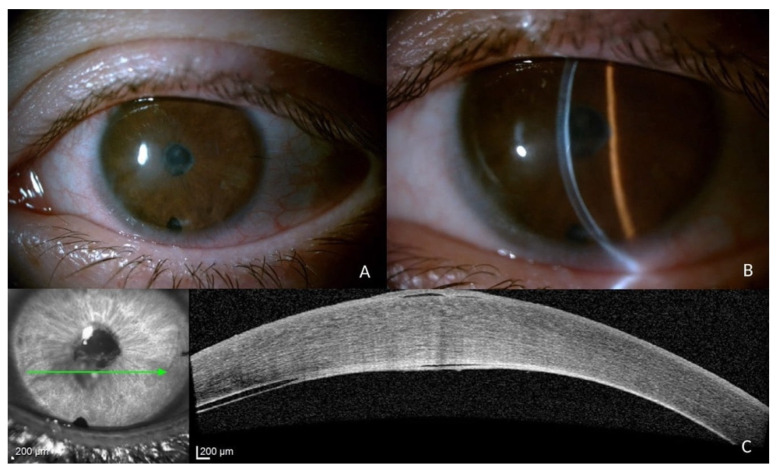
(**A**) Slit-lamp image showing significant fibrotic changes at the posterior corneal stroma–Descemet membrane interface and an inferior iridotomy; (**B**) the thin beam of the slit-lamp shows the complete adhesion of Descemet’s membrane; (**C**) AS-OCT after the second intracameral injection of SF_6_ 20% showing the definite resolution of the Descemet membrane detachment.

**Figure 3 jcm-12-00330-f003:**
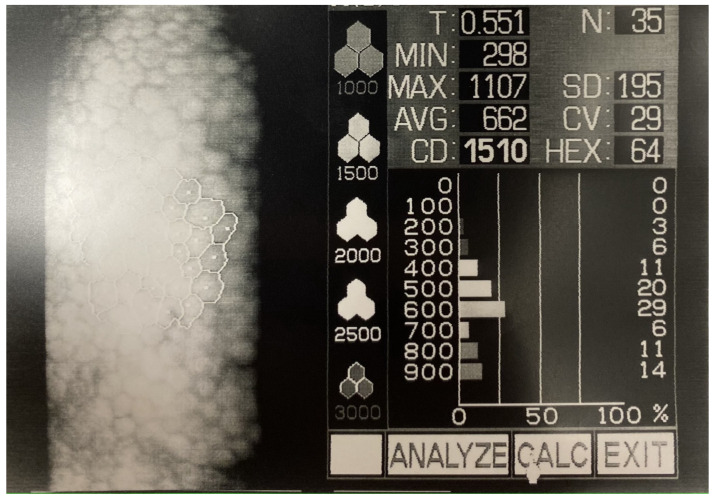
Endothelial cell count shows a low endothelial cell count after the episode of uveitis and the Descemet membrane detachment.

## Data Availability

The data presented in this study are available on request from the corresponding author.

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
