# Peer review of "Case Report: “Spontaneous Descemet Membrane Detachment”"

_jcm, 2022, doi:10.3390/jcm12010330_

Round 1

Reviewer 1 Report

Dear Authors,

In the study, which is valuable in terms of drawing attention to the rarely seen spontaneous DMD issue, it would be appropriate to have a more comprehensive conclusion sentence

Author Response

Thank you for the suggest, we added a final sentence to the end of the manuscript at line 227

Reviewer 2 Report

This manuscript reported a case and discuss the clinical characteristics and treatment of spontaneous Descemet membrane detachment, intracameral injection of 20% sulphur hexafluoride (SF 6 ) with complete resolution of the DMD.   This is valuable for early diagnosis and trentment of DMD to get good outcome. 

From the pictures of AS-OCT ,which indicating that the DM is rigidy,  there shold be a long time before the patient were refered to the authors, thus when did the cornea edema was reported by the docors previously should be put in the manuscript. 

As for  the result, the BCVA after the cure of the DCD should also be reported.

Author Response

Thank you for the observation, we added these informations to the manuscript at line 63 and 82